# Functionalization of Carbon Nanotubes Surface by Aryl Groups: A Review

**DOI:** 10.3390/nano13101630

**Published:** 2023-05-13

**Authors:** Pavel Oskin, Iraida Demkina, Elena Dmitrieva, Sergey Alferov

**Affiliations:** 1Laboratory of Ecological and Medical Biotechnology, Tula State University, Friedrich Engels Street 157, 300012 Tula, Russia; pavelfraj@yandex.ru; 2Chemistry Department, Tula State University, Pr. Lenina 92, 300012 Tula, Russia; 3Biotechnology Department, Tula State University, Pr. Lenina 92, 300012 Tula, Russia

**Keywords:** carbon nanomaterials, semiconductor nanotubes, metal nanotubes, carbon nanotube functionalization, Gomberg–Bachmann reaction, Billups reaction

## Abstract

The review is devoted to the methods of introducing aryl functional groups to the CNT surface. Arylated nanotubes are characterized by extended solubility, and are widely used in photoelectronics, semiconductor technology, and bioelectrocatalysis. The main emphasis is on arylation methods according to the radical mechanism, such as the Gomberg–Bachmann and Billups reactions, and the decomposition of peroxides. At the same time, less common approaches are also considered. For each of the described reactions, a mechanism is presented in the context of the effect on the properties of functionalized nanotubes and their application. As a result, this will allow us to choose the optimal modification method for specific practical tasks.

## 1. Introduction

Carbon nanotubes (CNTs), a present-day material discovered in 1993 [1], are widely used in various fields of human activity. Numbers of different CNTs forms have been obtained [2]. CNTs division is of the utmost interest, that depends on the number of layers (single-walled, double-walled, and multi-walled), as well as on the structure type (armchair”-like and zigzag), which affects the conductivity type (metallic and semiconductor, respectively) [3,4,5]. These types of nanotubes are the most notable for their properties. Due to the strong van der Waals interaction between aromatic systems, CNTs “stick together” into dense aggregates, making them almost insoluble, thus it significantly complicates the composite materials generation based on them. The use of various methods of CNT surface functionalization has been proposed to solve this problem, thereby weakening intermolecular interactions. The development of CNT functionalization methods has also enabled the extension of the application area due to a greater variety of CNT properties. A number of reviews [6,7,8,9,10,11,12,13,14,15,16,17,18] present data on this topic; however, most of them describe the CNTs’ chemistry generally. Although several reviews on specific methods of CNT functionalization have been published over the last few years [14,18,19,20,21], not all sections of CNT chemistry have been covered in sufficient detailed [22,23,24]. Moreover, a number of new studies require analysis and reflection.

The purpose of this work is a critical analysis of the CNT surface functionalization methods by aryl fragments to determine their effect on the properties of modified nanotubes. This reaction class has been chosen due to the wide application areas of arylated nanotubes, both by themselves and for further modification. At the beginning, various methods of arylation and their mechanisms are described in the context of the influence of test conditions on the structure and properties of functionalized CNTs. This is followed by a summary of the properties of nanotubes functionalized by aryl fragments, and their application.

## 2. Methods of Introducing Aryl Fragments to the CNT Surface

### 2.1. The Gomberg–Bachmann Reaction

Arylation by the Gomberg–Bachmann reaction with diazonium salts (Figure 1) is the most common method of modifying the surface of carbon nanomaterials. There is a number of publications on this topic [20,21,22,23]; therefore, we do not intend to compile an exhaustive review of it. This article emphasizes the data analysis of the reaction mechanism and various CNT modification methodologies since these aspects play an important role in the practical application of the Gomberg–Bachmann nanotube functionalization.

Diazonium salts are unstable in solution and extremely explosive in dry form, excluding some complex fluorides, aryl sulfonates [25], and o-benzene disulfimides [26]. Aqueous solutions of arylenediazonium borofluorides are most commonly used for the arylation of CNTs. Their functionalization is carried out either electrochemically [27] or due to the spontaneous decomposition of diazonium cations [28]. However, this approach has a number of disadvantages. CNTs are dispersed in water with the use of SDS, which certainly bind to nanotubes due to the strong hydrophobic interactions, and this is not always acceptable in a number of areas, such as electrochemistry and bioelectrocatalysis. Furthermore, arylenediazonium borofluorides are poorly soluble in water.

The production of arylenediazonium in non-aqueous media in situ is also widely used, this is achieved by the reaction between isoamyl nitrite and aromatic amine. However, this approach is not environmentally friendly, as well as the dispersibility of native nanotubes in organic solvents does not meet the requirements. The reaction has been proposed to be carried out in a two-phase water-isoamyl nitrite system [29], but this approach does not solve most of the problems.

The study [30] suggests a Gomberg CNT functionalization without the solvents. The technology is eco-friendly and can be easily scaled for the industrial use. Arylation takes place in a mixture of amine and isoamyl nitrite when heated to 60 degrees and extensively stirred. At the same time, CNTs are exfoliated from their aggregates by the mixing and under the influence of aryl radicals. Alkyl nitrite can be substituted with sodium nitrite in the presence of acid. These are CNT bundles that undergo functionalization, but not particular nanotubes. Nevertheless, solubility after functionalization increases significantly.

Triazenes are proposed to be used as an alternative to arylenediazonium borofluorides due to their greater stability and solubility in water [31]; however, the use of surfactants is also necessary to create CNT dispersion. Alternatively, a solid-phase reaction can be considered between CNT salts obtained by fusion with metallic potassium and arylenediazonium borofluorides. It is technically difficult and does not solve the instability problems of some borofluorides, although it allows increasing the selectivity of nanotubes with metallic conductivity [32].

The study [33] has shown that mechanochemical arylation of CNTs by Gomberg–Bachmann is possible by grinding nanotubes with arylenediazonium salts in the presence of ionic liquids based on imidazolium and potassium carbonate at room temperature for several minutes. The reaction is environmentally friendly, while the nature of the arylenediazonium cation and its counterion does not affect the degree of functionalization. It seems most promising to carry out arylation using a urea melt as a solvent, in which nanotubes are perfectly soluble without exposure to ultrasound, and diazonium cations can be obtained by the interaction of aromatic amine with sodium nitrite [34,35,36,37]. This approach is also highly environmentally friendly.

There is still no consensus on the detailed mechanism even for the classical Gomberg–Bachmann reaction; nevertheless, there have been attempts to describe the mechanism of CNT arylation, mainly for aqueous CNT dispersions in the presence of surfactants at different pH values [38,39,40,41,42]. The radical nature of the mechanism has been confirmed by the fractional order of the reaction, and the ability of free radical traps to stop the reaction [39]. The formation of an aryl radical from a diazonium cation can occur either due to spontaneous decomposition or due to reduction by a carbon nanotube (Figure 2) [39]. The first mechanism is typical for an alkaline medium, as confirmed by the absence of nucleophilic addition on the CNT surface [38]. The authors of [33] have proposed the second mechanism for an acidic medium, but there is no direct evidence of the formation of CNT cation radicals. The EPR spectrum signal from radicals with high electron delocalization can be correlated with aryl-CNT radicals formed at the next stage. At the same time, this study shows that metal nanotubes play the role of catalysts, due to the active reduction of arylenediazonium cations. There is no data about a side effect of the nucleophilic addition of fluoride or hydroxide ions, which should actively proceed with nanotube cations. This fact raises many questions.

The occurring aryl radicals attach to nanotubes, reacting as electrophiles. Due to this fact, there can be observed both the selectivity to CNTs with a metallic type of conductivity and a higher reaction rate of aryl radicals with acceptor substituents [39]. The resulting aryl-CNT radicals react with diazonium salt, while the structure is stabilized. In the acidic media, a nucleophile from the solution should be attached to the CNT; however, no data confirming this could be found in the literature (Figure 3). The aryl fragment is split off from the nanotube if there occurs no addition of the diazonium cation to the aryl-CNT radical. This has been shown by measuring the electrical conductivity of nanotubes during the reaction. Electrical conductivity increases after the termination of the reaction, which is explained by the cleavage of a part of the aryl groups [40].

Studies [41,43], based on Raman spectroscopy data, state that the selectivity of Gomberg arylation is due to the formation of an intermediate with charge transfer from the arylenediazonium cation to the nanotube. Schmidt et al. [39] have shown that free radical traps stop the functionalization of nanotubes, which indicates the presence of phenyl radicals in free form. Further, the formation of a charge transfer complex contradicts the mechanism determined by GC-MS and X-ray fluorescence analysis [38].

Adding diazonium cations in small portions has been suggested to achieve a high degree of arylation selectivity [43]. The attachment of a radical to the CNT surface facilitates the arylation of neighboring positions, due to which the selectivity to metal nanotubes increases. The study [44] confirms this fact and shows that the selectivity completely disappears at a high concentration of the diazonium cation in solution.

The Gomberg–Bachmann reaction shows selectivity both to the type of CNT conductivity, and to their diameter, but the data is contradictory. Studies [41,45] report that arylation proceeds more easily for CNTs of smaller diameter, while the publications [39,42] observe the opposite effect. The authors of [39] explain these differences by using surfactants of different natures. This hypothesis requires further studying and confirmation. It should be also noted that long chains of benzene rings are attached due to the radical reaction during the arylation of CNTs [46].

Information about side reactions is also contradictory. On the one hand, GC-MS analysis of the filtrate has not revealed the presence of polynuclear aromatic hydrocarbons [38]. On the other hand, the study [47] has treated Gomberg–Bachmann arylated nanotubes with repeated washing by different solvents and has observed the fluorescence of washing waters due to by-products of the reaction. Considering the adopted to-date reaction mechanism, there are no theoretical prerequisites for the absence of side processes, unlike the Billups reaction discussed below. Further research is required to give a decisive answer to the question of the purity of Gomberg–Bachmann functionalized CNTs.

### 2.2. Billups Reaction

The reductive arylation of CNT salts with various reagents (the Billups reaction and its modifications) is of great interest. It has been shown since 1997 that an electron is transferred from a metal atom to carbon when CNTs are fused with alkali metals [48]. Due to Coulomb repulsion, CNT salts with alkali metals have high spontaneous solubility—up to 2 mg/g in DMSO and DMF, and up to 4 mg/g in sulfolane; moreover, the solutions are stable for up to a year and resemble properties of polyelectrolytes in an inert atmosphere [49,50]. The addition of dibenzo-18-crown-6 increases solubility, due to complexation with sodium ions [51]. Further, there is no need to use ultrasound, which destroys the structure, when preparing solutions of CNT salts [52,53,54,55].

CNT salts can be obtained by several methods (Figure 4), interaction with an alkali metal solution in liquid ammonia is the most common [52,53,56,57,58,59,60,61,62,63,64,65,66,67,68,69,70], and lithium is mostly used. Alkali metal is embedded inside CNTs, lithium can also form covalent bonds with carbon atoms [69], and alkali metal is possible to be embedded between the layers of tubes with subsequent arylation/hydrogenation of the inner tubes in the case of CNTS [71]. The functionalization degree may be controlled by maintaining an optimal metal-carbon ratio [72]. The study [56] has shown that the nature of the alkali metal does not affect the course of arylation; however, the research [52] states that the degree of functionalization, determined thermogravimetrically, increases in the Na-Li-K series. Furthermore, the addition ratio between 1,4 and 1,2 differs [53]. There is not enough research on this issue, so it requires further studying.

The reaction in THF with alkali metals in the presence of electron–carrying catalysts, such as naphthalene [61,72,73,74,75,76], benzophenone [74], or 4,4’-di-tert-butylbiphenyl is another commonly used method of obtaining CNT salts [77]. However, the reaction product turns out to be a contaminated catalyst since it is difficult to remove it due to the strong stack interactions with CNT. This can lead to deterioration of the electrochemical properties of CNTs. Several studies use butyllithium or another lithium alkylide in cyclohexane or THF to produce CNT salts [66,78,79,80,81,82,83,84]. However, in this case, alkyl radicals are also sewn onto the surface of CNTs in addition to aryl ones. The fusion of CNTs with alkali metals in an argon atmosphere [32,55,85,86] or interaction with sodium amalgam in toluene in the presence of dibenzo-18-crown-6 is less common due to the technical complexity [51]. It has also been proposed to replace liquid ammonia with ethylenediamine, but this approach has not been widely used [58,87]. The mechanochemical modification of CNT [88] is of interest since this method can be easily scaled. CNT, metallic potassium, chlorobenzene, and a ball are placed in a steel capsule, then the mixture is mixed in a planar mill in a nitrogen atmosphere. Excess potassium is removed by reaction with isopropanol. The functionalized nanotubes have increased solubility in chloroform, methylene chloride, toluene, and dichlorobenzene.

Aryliodides are mostly used for the arylation of CNT salts [55,56,57,72,73]; therefore, their interaction with CNT has been studied in full detail. Thus, it is shown that the degree of functionalization is higher if there are donor substituents in the aryl radical [56]. Reducing arylation proceeds more easily for nanotubes of smaller diameters [62]. The study [65] proposes a reactor to produce alkylated and arylated nanotubes by the Billups–Birch reaction on a semi-industrial scale. It has been shown [63] that careful mixing of the reaction mixture is necessary to obtain reproducible results during synthesis in this reactor. Strong cooling slows down the formation of lithium amide, which leads to more accurate compliance with the stoichiometric lithium–carbon ratio, thereby increasing the reproducibility of the properties of modified CNTs, as well as the degree of their functionalization. Removal of the residual amount of iron catalyst, for example, by chlorination, leads to an increase in the functionalization degree, since iron catalyzes the decomposition of solvated electrons.

The synthesis of electrically conductive polymers is proposed by the interaction of CNT salts with paradiodobenzene and paradiodobiphenyl, alongside benzene-4,4′-bis (diazonium) and 1,1′-biphenyl-4,4′-bis (diazonium) [55] (Figure 5). The most effective functionalization occurs using aryliodides as regards the volume of homogeneous functionalization and the functionalization degree. Phenylene linkers give a greater functionalization degree, while biphenyl linkers provide a larger surface area and improved electrochemical properties. The maximum degree of functionalization has been achieved with a stoichiometric ratio of potassium: carbon 1:4.

Using aryliodides as arylating agents leads to the fact that functionalized CNTs turn out to be contaminated by-products of a combination of radicals that are difficult to remove. This has been shown by analysis of GC-MS filtrate after alkylation of CNT [52,59]. Alkylation is carried out with other reagents (Figure 6), such as peroxides [52,64,74], cyclic halides [64], sulfides and disulfides [68], carbonyl compounds [66], acetylenes [80], diazonium salts [32,55,89] and iodonium [76] to solve this problem. The free radicals formed from these compounds give unstable combination products that immediately disintegrate. This is proved by GC-MS analysis of the filtrate regarding cyclic halides [64]. The use of carbonyl compounds is of particular interest. Unlike peroxides, sulfides, and cyclic halides, they are more accessible and more reactive than aromatic acetylenes. The reaction cannot be carried out in liquid ammonia, since the carbonyl group is reduced in this case. The activity of carbonyl compounds in this reaction depends on the stability of the intermediate-formed carbocation [66]. The disadvantages of this approach include a low functionalization degree.

The common disadvantage of alkylation and arylation reactions by Billups is a side hydrogenation process, occurring during both the reaction [58] and the decomposition of the reaction mixture with ethanol [58,71,73,90]. The hydrogenation degree depends on the used solvent and decreases in the set of liquid ammonia > ethylenediamine > THF [90]. The use of sodium amalgam in the crown ether for the synthesis of CNT salts reduces the hydrogenation degree even more. This approach is difficult to scale, and the functionalization degree is lower than other methods [56]. The use of ethylenediamine as a solvent is of the greatest interest, since there is no contamination of functionalized CNTs with catalysts in this case, as in the synthesis of CNT salts in THF and hydrogenation occurs to a lesser extent than in liquid ammonia.

### 2.3. Reactions with Peroxides and Related Compounds

Thermal decomposition of benzoyl peroxide is widely used for CNT phenylation [91,92,93,94,95,96]. The reaction has been carried out in boiling benzene (o-chlorobenzene) [91,92,94] in an argon atmosphere or boiling toluene [93] in the air. Despite the higher boiling point of toluene and the same ratio of reagents, the study [93] shows a lower functionalization degree than others (Table 1). This results from the fact that oxygen reduces the concentration of free radicals formed during the decomposition of benzoyl peroxide.

A solid-phase reaction is also described [96]. Other peroxides can be used in addition to benzoyl peroxides, such as p-methoxybenzoyl peroxide or phthaloyl peroxide [92]. Peroxides can also be used as initiators of radical reactions with other compounds, for example, aryl and alkyl iodides [91]. Phenylated nanotubes can also undergo further functionalization, e.g., sulfonation in oleum to increase their solubility in water [94].

The mechanism of CNTs’ interaction with organic peroxides has been studied in several studies [92,93,95]. Engels et al. [91] have shown that thermolysis of peroxides occurs faster in the presence of CNTs, and catalysis occurs due to interaction with CNTs but not because of the trace amounts of iron remaining after the nanotube synthesis. A similar effect is observed when benzoyl peroxide is decomposed in ethanol at room temperature [95]. The analysis of the ethanol solution carried out after the completion of the reaction by the GC-MS method has not revealed the products of phenyl radicals’ combinations with each other or with a solvent. The formation of free radicals from carbon nanotubes has also been proved by the EPR method. The mechanism explaining these processes is shown in Figure 7 [92].

CNTs reduce peroxides to radical anions, which immediately undergo the breaking of O-O bonds. The radicals rapidly react with the cation radicals of the nanotubes after decarboxylation, and functionalization occurs. In the study [93], the PMR method has shown that phenyl radicals are attached to CNTs; thus, benzoyl peroxide first cleaves off carbon dioxide, and only then reacts with CNTs. At the same time, the photoluminescence of arylated nanotubes has determined the presence of phenyl groups attached via oxygen bridges [97]. Photoluminescence is much more sensitive than PMR spectroscopy; therefore, phenyl fragments, sewn to CNTs through an oxygen bridge, are probably negligible.

Reactions of CNT arylation by structural analogs of peroxides have been described, which apparently proceed by a similar mechanism. Phenyl radicals have been sewn through a sulfide bridge by boiling CNTs with a twofold excess of organic disulfide in toluene for 48 h without oxygen [98]. The degree of functionalization has reached 1 functional group per 39 carbon atoms, which is 2 times lower for a similar reaction with benzoyl peroxide. Thus, disulfides are less reactive arylating agents, meanwhile, they enable the introduction of sulfur-containing functional groups onto the surface of nanotubes. This specific feature can play an important role both in the further functionalization of CNTs and in the immobilization of gold or silver nanoparticles.

Wong et al. [99] have shown that CNT can be joined by radicals formed from hexaphenyldisilane under the action of mercury-xenon lamp light. The reaction is selective to CNTs of small diameter with semiconductor conductivity.

### 2.4. Other Arylation Methods

There is a report on the use of the Ullmann reaction for the arylation of CNTs [100]. Initially, CNT has been chlorinated by reacting with iodine trichloride in tetrachlorocarbon for 3 hours. Chlorinated CNTs have reacted with iodobenzene, phenol, or aniline in the presence of cesium carbonate, copper chloride 1, and phenanthroline in DMF, at 120 degrees for 2 days. The functionalization degree has reached 3.5 mmol/g. It is shown that functionalization proceeds best for CNTs of smaller diameters.

CNT has been proposed to be arylated with 4-methoxyphenylhydrazine hydrochloride by boiling in toluene in an oxygen medium [101] or irradiating with microwave radiation. This turns out to be more effective [102]. Arylated CNTs are better soluble in o-chlorobenzene. CNT modification with phenylhydrazine by interaction in an aqueous solution in the presence of SDS for 2 days has also been described [103]. Covalent crosslinking with CNT walls is proved by Raman spectroscopy. The authors do not propose a reaction scheme or any product structure. Solubility has increased in modified CNTs, elemental analysis has shown nitrogen concentration growth.

The study [104] shows the possibility of introducing triphenylphosphine groups to the CNT surface. Initially, the Gomberg–Bachmann arylation of CNTs with p-bromaniline has been performed. Then, the interaction with lithium diphenylphosphide has been carried out. Various physicochemical methods characterize the product. It is shown that a lot of carbon is formed during thermal decomposition, which can be used in extinguishing fires.

### 2.5. Alternative Methods of Introducing Aryl Groups to the CNT Surface

The cross-linking of aromatic hydrocarbons with nanotubes via a carboxamide bond and cycloaddition reactions do not belong to traditional arylation methods, nevertheless, they allow introducing aromatic fragments onto the CNT surface. Therefore, they can be used to solve the same problems. The production of aromatic amides from CNTs has a number of disadvantages: the multi-stage nature of the process, the strong destruction of the CNT structure, as well as the crosslinking of aryl groups mainly with edge carbon atoms. This characteristic does not allow to get a high functionalization degree; thus, the main focus is paid to cycloaddition reactions in this section (Figure 8). There has been a short review published on them [13]. This approach to the functionalization of the nanotube surface is used less commonly than others, due to the inaccessibility of precursors, though there are exceptions: the cycloaddition of anthracene or dimethylanthracene [99,100].

The studies [105,106] have modified CNT with a mixture of aldehyde and a primary amine, from which a Schiff base has been formed directly in the reaction mixture, being attached to CNT by the 1,3-cycloaddition reaction. Functionalized CNTs are highly soluble in chloroform, methylene chloride, acetone, methanol, ethanol, and water, less soluble in toluene and THF, and practically insoluble in less polar solvents, including diethyl ether and hexane. The solubility in chloroform has reached 50 mg/mL without ultrasound treatment. The functionalization degree has been estimated by the absorption spectra in the UV region, it has represented 1 group per 95 carbon atoms.

The Diels–Alder cycloaddition is also described for CNT. Nanotubes can act both as dienes and as dienophiles [107]. However, their activity is low in these roles; therefore, it is necessary either to introduce acceptor groups by fluorination or oxidation [108,109], or to use active reagents for the reaction [109,110]. The authors of [111] propose to carry out the reaction at elevated pressure in the presence of chromium hexacarbonyl, which increases the activity of dienes. Single-walled carbon nanotubes are more active in the Diels–Alder reaction than multi-walled ones, which is due to the high annular deformation of single-walled nanotubes [110]. The disadvantages of this approach include high energy costs and contamination of functionalized CNTs with chromium compounds. It is important to note that the Diels–Alder cycloaddition is reversible [107].

The study [112] proposes to carry out the cycloprecoupling of benzene cyclobutene to CNT. The reaction allows to control the functionalization degree by selecting the optimum. Further, there is no need to use high pressure, catalysts, as well as pre-functionalized CNTs. The disadvantages include the inaccessibility of precursors.

### 2.6. Fractionation of Carbon Nanotubes

All scalable methods of CNT synthesis give a mixture with a wide range of properties: length, diameter, and type of conductivity. The isolation of individual CNT fractions is important for the most effective use of their extraordinary properties. The most acute problem is the separation of metal nanotubes from semiconductor ones [113]. Semiconductor CNTs are used to create transistors [114]; moreover, they have photoluminescence [115,116,117,118]. The admixture of metal CNTs worsens their properties significantly, causing incorrect transistors’ functioning. Gomberg–Bachmann arylation proceeds selectively to metallic nanotubes [43,44,119,120,121]. Selectivity may be lost when the arylenediazonium salt is over the limit. The optimal ratio is functional group: carbon 1:100–1:50 [44]. The introduction of various functional groups provides the separation of metallic and semiconductor nanotubes based on the difference in solubility [120], electrophoretic mobility [119,121], or other physicochemical properties. It has been proposed to arylate CNT salts to increase the selectivity of the reaction [32]. It is important to note that the arylation reactions are reversible [43,67], which allows the restoration of the original structure of nanotubes after their fractionation.

It is also possible to fractionate nanotubes by diameter with the help of arylation. Thus, the Billups reaction [62] and arylation with peroxides [93] or disilanes [99] are mainly carried out in CNTs of a smaller diameter, while the 1,3-cycloaddition of pyridinium ylides proceeds more easily in CNTs of a larger diameter [122].

## 3. Properties and Application of Aryl-Group Functionalized CNTs

The properties of modified CNTs are closely related to the functionalization method in use, so it is necessary to choose a special approach for each application area. Table 2 summarizes the data described in this work, allowing us to choose the optimal method.

### 3.1. Solubility of Arylated Carbon Nanotubes

A dispersibility increase in different solvents is one of the most important properties of CNTs functionalized by aryl groups. Reviews on this topic have recently been published [18,123]. The solubility of functionalized CNTs is not quantified in most studies. As usual, the authors show that native nanotubes, generally, are not capable of forming stable dispersions in a particular solvent. The research [30] shows that the CNT solubility differs from one author to another due to the use of different methods for its determination. Nevertheless, we have tried to evaluate the effect of the CNT arylation on their solubility (Table 3). In all cases, except [94], filtration of a stable solution with subsequent drying and weighing of the filter has been used to determine the solubility. In [94], the concentration of CNTs in an aqueous solution has been determined spectrophotometrically.

Despite the low functionalization degree (1:69–1:272), CNTs, functionalized according to Billups, have the greatest solubility [57]. Most likely, this is due to the fact that a stable CNT dispersion is formed due to Coulomb repulsion between particles while the functionalization. Sewn to the surface, aryl groups prevent the subsequent adhesion of CNTs after the nanotubes return to the uncharged form. Other methods do not increase CNT solubility so effectively since the functionalization occurs mainly on the surface of the nanotube ropes. Using an aqueous SDS solution for CNT dispersion is apparently less effective than obtaining nanotube salts.

### 3.2. Photoluminescence of Arylated Carbon Nanotubes

It is widely known that semiconductor carbon nanotubes are capable of photoluminescence in the near IR range [115,116,117,118]. Arylation makes it possible to introduce small defects in the CNT structure in a targeted manner, which makes it possible to achieve more intense luminescence [115,116,117,127,128,129,130,131,132,133,134,135,136,137,138,139,140]. Further, unlike alkylation and oxygen doping, arylation provides more opportunities to influence the luminescence spectrum: the use of aryl radicals of different structures [127,128,131,138,140], subsequent modification of aryl radicals [135,138], selective arylation under the action of light [137,139], biarylation with different bridge lengths between sewn aryl groups [136]. It is important to note that arylated CNTs have the rare property of luminescent solvatochromism, which may have unusual applications in the future [130]. A photoluminescent sensor has been developed for local determination of pH with an accuracy of 0.2 units [129], as well as a sensor for selective determination of metal ions [135], based on arylated semiconductor CNTs.

### 3.3. Development of Bioelectrodes with Surface-Oriented Immobilization of Enzymes

Bioelectrocatalysis plays an important role in modern chemistry [141]. The creation of electrodes with oriented immobilized enzymes is necessary for the development of biofuel cells [47,142,143,144,145,146,147,148,149,150,151,152], hybrid batteries [144,153,154,155,156,157], and biosensors [158]. CNTs are a promising basis for the creation of bioelectrodes due to their high electrical conductivity and surface area. Enzymes having hydrophobic pockets, such as laccase, bilirubin oxidase, and fructose dehydrogenase, can bind to aryl radicals on the electrode surface, which facilitates electron transfer. Aryl groups are often introduced through the acylation of aromatic amines with oxidized CNTs [47,143,153,159,160,161] or Gomberg–Bachmann arylation [47,142,147,152,153,154,162,163]. A non-covalent modification of CNT by pyrene derivatives has also been proposed [150,164,165]. Comparing the effectiveness of these functionalization methods is difficult due to the small amount of information and its inconsistency. Thus, the study [47] states that the terephenyl radical binds to laccase at its worst, whereas the research [143] shows the opposite. Further, each work uses enzymes isolated from their own type of microorganisms, which also makes it difficult to compare. However, anthracene arylated nanotubes are used most of all [144,145,146,147,148,149,150,151,152,153,155,156,158,160,161,166].

## 4. Conclusions

Many methods of introducing aryl groups to their surface have been described since the discovery of CNTs, but a more detailed study of the functionalization mechanisms is needed, as there are only a few works devoted to this problem and there appear to be serious contradictions. At the same time, understanding the reaction mechanism will make it possible to obtain materials with specified properties, which is the ultimate goal. All the approaches discussed in this review have their advantages and disadvantages (see Table 2), so they can be used in different industries, but additional research is required for their effective use. The work in [65], using the example of the Billups reaction, when CNT functionalization is scaled to industrial scales to obtain reproducible results, shows the necessity to consider many subtleties that are invisible in laboratory synthesis conditions. Most of the presented approaches are based on radical reactions; however, the use of the chemistry of organometallic compounds, which gives unusual results, is of considerable interest. Based on the foregoing, for each area of practical application, a specific type of functionalization can be recommended. For fractionation of CNTs according to the conductivity, it is best to use the Gomberg–Bachmann reaction; the highest increase in solubility gives the Billups reaction followed by sulfonation; functionalization with peroxides is most effective for activation of photoluminescent properties. Concerning the oriented enzyme immobilization for bioelectrocatalysis, the data are contradictory.

## Figures and Tables

**Figure 1 nanomaterials-13-01630-f001:**
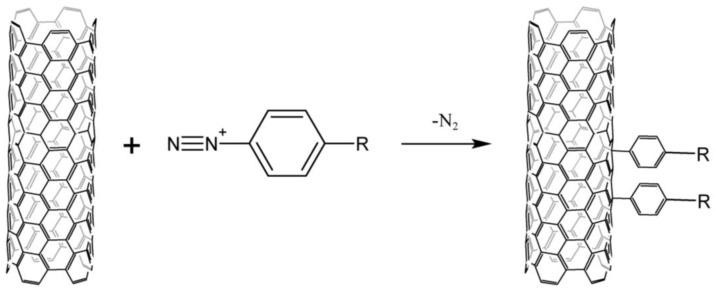
Scheme of CNTs arylation by the Gomberg–Bachmann reaction with diazonium salts.

**Figure 2 nanomaterials-13-01630-f002:**
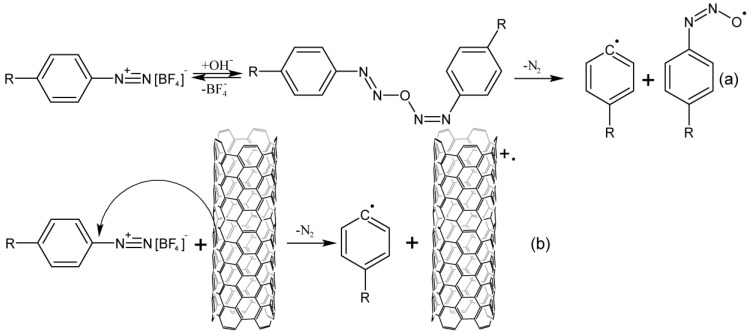
The formation of aryl radicals from arendiazonium salts: (**a**)—decomposition of diazonium salts in an alkaline medium. (**b**)—a one–electron reduction of the CNT diazonium cation by the type of the Zandmeir reaction in acidic media.

**Figure 3 nanomaterials-13-01630-f003:**
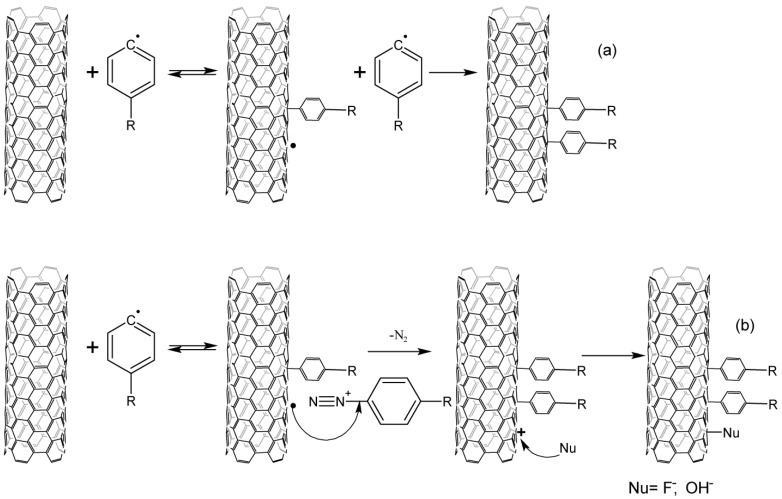
Interaction of CNTs with aryl radicals and arylenediazonium cations, (**a**)—alkaline media, (**b**)—acidic media.

**Figure 4 nanomaterials-13-01630-f004:**
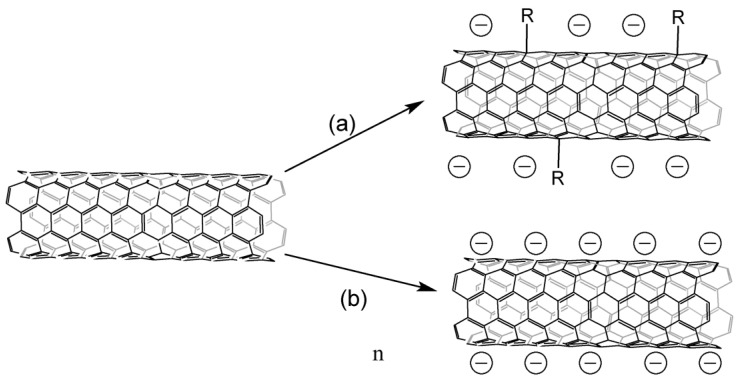
Obtaining CNT salts: (**a**)—interaction with lithium alkylides, (**b**)—interaction with alkali metals in solvents of various nature in the presence or absence of electronic transport mediators.

**Figure 5 nanomaterials-13-01630-f005:**
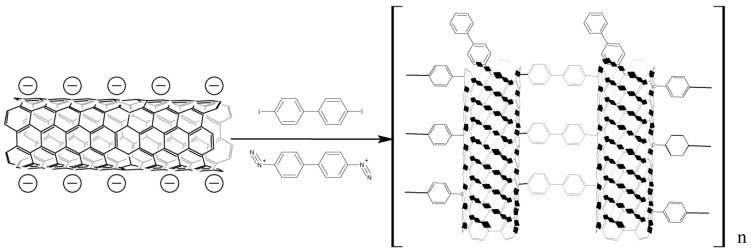
Synthesis of conductive polymers based on CNT salts and arendiazonium salts or aryliodides.

**Figure 6 nanomaterials-13-01630-f006:**
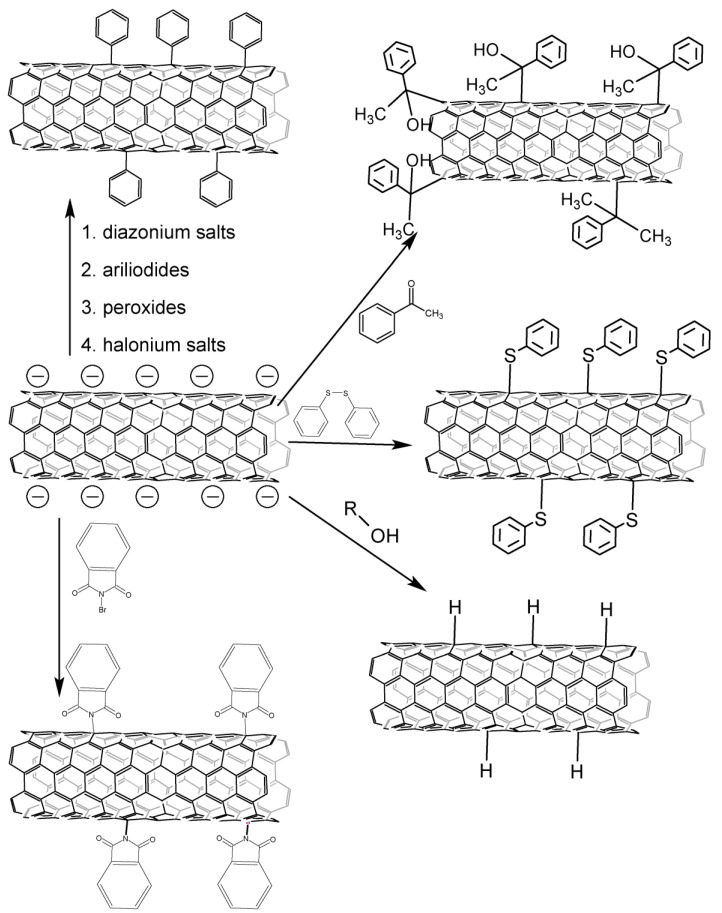
Arylation of CNT salts with various reagents.

**Figure 7 nanomaterials-13-01630-f007:**
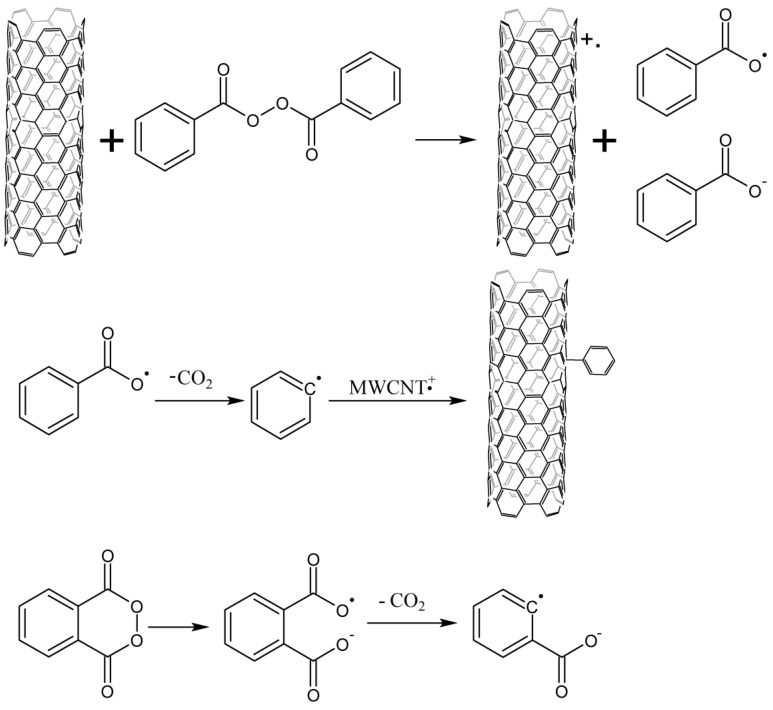
Mechanism of CNT arylation by peroxides (formation of radicals from cyclic and acyclic peroxide).

**Figure 8 nanomaterials-13-01630-f008:**
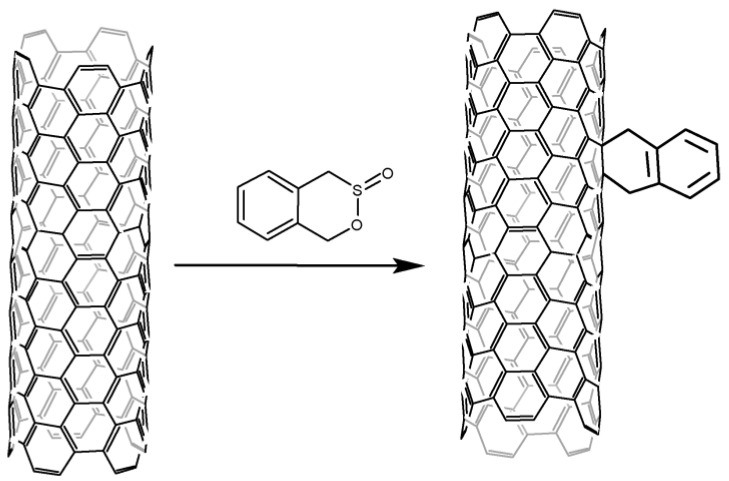
Cycloaddition to CNT under microwave irradiation.

**Table 1 nanomaterials-13-01630-t001:** Effects of reaction conditions on the degree of CNT functionalization.

Peroxide	Solvent	Atmosphere	Time (Hours)	Mole Ratio (C:peroxide)	Functionalization Degree (C:group)	Ref.
benzoyl peroxide	toluene	air	9	1:0,83	1:36	[93]
benzoyl peroxide	benzol	argon	24	1:1	1:14	[91]
benzoyl peroxide	benzol	argon	2	1:1	1:18	[94]

**Table 2 nanomaterials-13-01630-t002:** Application areas of various methods of CNT surface functionalization with aryl groups.

Reaction	Mechanism	Advantages	Disadvantages	Properties and Application
Gomberg–Bachmann reaction	Described for aqueous dispersions of CNTs. Radical mechanism in the alkaline media, cationic-radical mechanism in the acidic media. Direct evidences for the formation of CNT radical cations is currently lacking. Additional studies of the mechanism in non-aqueous media are required.	The availability of arylating agents, the presence of “green” methods of arylation, a wide range of functional groups that can be introduced on the CNT surface, selectivity to different types of CNTs.	The possible presence of by-products of the radical’s combination, there is no consensus on this. It is possible to use electrochemical reduction of diazonium salts to guarantee the elimination of adverse reactions. However, this approach is difficult to scale. Explosion hazard of reagents.	Functionalized CNTs have increased solubility both in water and in non-aqueous media, and are used to immobilize enzymes for biofuel cells and biosensors. The selectivity of the reaction with respect to nanotubes with different types of conductivity allows them to be fractionated. Activation of CNT photoluminescence by Gomberg–Bachmann arylation is also possible.
Billups reaction	The mechanism of the anion-radical reaction is well developed. There are many options for both the preparation of CNTs radical anions and the reagents used for their functionalization. Data on the influence of the alkali metal nature on the course of the reaction are contradictory, and additional studies are required.	High selectivity and absence of radical combination products, the ability to control the degree of functionalization, scalability to an industrial scale, a wide range of functional groups available for introduction to the CNT surface.	A side reaction of hydrogenation, which can be minimized by the selection of a solvent and the conditions of the reaction. The use of liquid ammonia or amines as solvents. Alternative approaches have been proposed, but they have their disadvantages. For example, alloying with alkali metals requires an inert atmosphere and a chemically resistant reactor. High requirements for the purity of CNTs.	Functionalized nanotubes are well dispersed in non-aqueous media. The reaction is convenient to use for the synthesis of polymer-based CNT composites, including conductive ones, since anion radicals are the initiators of polymerization. It is possible to use the reaction for fractionation of CNTs by diameter.
Arylation with peroxides	The mechanism is radical. Aryl radicals are mainly attached directly to the CNT surface; however, attachment via oxygen bridges is also possible.	The simplicity of the reaction, scalability, the possibility of introducing a small number of aryl groups, the absence of side reactions of a combination of free radicals with each other or with a solvent.	A small range of functional groups available for introduction to the CNT surface. Explosion hazard of reagents.	The reaction is used to increase the solubility of CNTs, as well as to influence their photoluminescence. It is possible to use the reaction for fractionation of CNTs by diameter.
Ullmann reaction	The mechanism is radical. Poorly studied, there is only one publication on the topic.	Selectivity to nanotubes of smaller diameter.	The high cost of reagents and the complexity of the reaction.	Up to date, the reaction has only theoretical significance.
Arylation with phenylhydrazine derivatives	A radical mechanism is suggested, but there is no direct evidences.	Simplicity of reaction and availability of reagents.	Possible by-products of a phenyl radical combination.	Functionalization promotes the growth of CNT solubility.
Diels–Alder reaction	4 + 2 cycloaddition	Ability to control the functionalization degree.	Requirements for the presence of acceptor groups on the CNT surface, or hard-to-reach reagents, or a catalyst and high pressure.	High solubility in non-aqueous media.
Amidation of oxidized CNTs	Nucleophilic substitution	Simplicity of synthesis, wide range of introduced functional groups, well developed methods.	Multistage, destruction of the CNT structure.	It is mainly used for the immobilization of enzymes in the development of a biofuel cells.

**Table 3 nanomaterials-13-01630-t003:** Effect of the CNT arylation method on the solubility.

CNT	Modification Method	Phenyl Radical	Solvent	S, * mg/mL	Ref.
single-walled	-	-	THF	0.005	[124]
single-walled	-	-	o-dichlorobenzene	0.095	[30]
single-walled, HiPco	Gomberg–Bachmann reaction, aqueous solution + DDS, decomposition at room temperature	p-tert butylphenyl	o-dichlorobenzene	0.7	[125]
			DMF	0.8	
			CHCl_3_	0.6	
			THF	0.6	
single-walled, gas-phase growth catalyst—iron	Gomberg–Bachmann reaction, aqueous solution, electrochemical reduction	p-tert butylphenyl	THF	0.05	[125]
single-walled, HiPco	Phenylation with benzoyl peroxide followed by oleum sulfonation	p-sulfophenyl	H_2_O	15	[94]
multi-wall, Bucky	Reductive acylation by Billups–Birch	p-sulfophenyl, ammonium salt	H_2_O	30	[57]
		p-sulfophenyl	H_2_O	15	
		p-isopropylphenyl	CHCl_3_	60	
		p-dodecylphenyl	CHCl_3_	100	
	Reducing acylation by Billups–Birch, lithium instead of sodium	p-dodecylphenyl	CHCl_3_	120	
Synthesis from carbon dioxide by laser ablation	Gomberg–Bachmann arylation, heating with isoamyl nitrite and aromatic amine in o-dichlorobenzene, 60 degrees	p-nitrophenyl	toluene	0.5	[126]
			DMF	0.7	
		p-carboxyphenyl	THF	0.6	
			CH_3_OH	0.6	
			CHCl_3_	0.8	
		p-butylphenyl	toluene	0.5	
			DMF	0.7	
single-walled	Gomberg–Bachmann arylation in urea melt	dimethylisophthalate	acetone	0.03	[36]
			H_2_O	0.03	
			DMF	0.09	
			ethanol	0.03	
		benzenesulfamide	acetone	0.09	
			H_2_O	0.05	
			DMF	0.09	
			ethanol	0.09	
		p-anisole	acetone	0.1	
			H_2_O	0.1	
			DMF	0.1	
			ethanol	0.1	

* Solubility.

## Data Availability

Not applicable.

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
