# Peer review of "Functionalization of Carbon Nanotubes Surface by Aryl Groups: A Review"

_nanomaterials, 2023, doi:10.3390/nano13101630_

Round 1

Reviewer 1 Report

The review paper is well developed, only a few remarks:

The authors indicated (lines 39-40) : The purpose of this work is a critical analysis of the CNT surface functionalization  methods by aryl fragments to determine their effect on the properties of modified nano-40 tubes. This purpose needs to be reforced along the paper, perhaps with quantitative data.

Lines 34-38. Include references for supporting the statement done.

Conclusion section should be more critical, guidelines are necessary for the reader.

Author Response

Manuscript ID: nanomaterials-2390043

Type of manuscript: Review

Title: Functionalization of Carbon Nanotubes Surface by Aryl Groups: A Review.

Authors: Pavel Oskin, Iraida Demkina, Elena Dmitrieva, Sergey Alferov *

Received: 24 April 2023

Dear Reviewer!

We would like to thank you for your constructive evaluation of our manuscript. We have completed a revision of the manuscript and have addressed all of the comments/issues raised by each Reviewer.

Any revisions to the manuscript were marked up using the «Track Changes» function. Please find our detailed response below:

Reviewer #1

The review paper is well developed, only a few remarks:

  1. The authors indicated (lines 39-40) : The purpose of this work is a critical analysis of the CNT surface functionalization methods by aryl fragments to determine their effect on the properties of modified nano-40 tubes. This purpose needs to be reforced along the paper, perhaps with quantitative data.

RESPONSE 1:

Table 3 has been added to the section 3 summarizing the results and application areas of various methods of CNT surface functionalization.

  1. Lines 34-38. Include references for supporting the statement done.

RESPONSE 2:

References [14,18-21] and [22-24] included.

  1. Conclusion section should be more critical, guidelines are necessary for the reader.

RESPONSE 3:

Conclusions have been revised.

Reviewer 2 Report

In the manuscript entitled "Functionalization of Carbon Nanotubes Surface by Aryl Groups: A Review" Alferov et al. review  the methods of introducing aryl functional groups onto CNTs surface mainly by means of Gomberg-Bachmann and Billups reactions, and the decomposition of peroxides. I believe that this review article is suitable for publication in Nanomaterials Journal after some revisions as indicated in the comments below.

line 60

repleace "comonly" with commonly

line 63

Please, define "SAS"

line 65-66

"Thus, surfactants interfere during the enzyme immobilization on the nanotube surface."

Why do the authors talk about an enzyme? please clarify

line 77

repleace "substitute" with substituted

Figure 3

Check the accuracy of the reaction mechanism depicted in Figure 3.

Line 128

"Homberg arylation" should be Gomberg arylation. This typos is repeated throughout the manuscript, check carefully.

line 159

repleace "DMFA" with DMF.

line 209

Replace "paradiodbenzene and paradiodbiphenyl" with paradiodobenzene and paradiodobiphenyl

Line 235

"Bilapse" should be Billups.

Table 1

Please, check the correspondence of the references cited in the table 1. Maybe there was an error in the numbering.

Figure 7

Check the accuracy of the reaction mechanism depicted in Figure 7.

Line 305

Replace "DDS" with SDS

Table 2

Please, check the correspondence of the references cited in the table 2. Maybe there was an error in the numbering.

Author Response

Manuscript ID: nanomaterials-2390043

Type of manuscript: Review

Title: Functionalization of Carbon Nanotubes Surface by Aryl Groups: A Review.

Authors: Pavel Oskin, Iraida Demkina, Elena Dmitrieva, Sergey Alferov *

Received: 24 April 2023

Dear Reviewer!

We would like to thank you for your constructive evaluation of our manuscript. We have completed a revision of the manuscript and have addressed all of the comments/issues raised by each Reviewer.

Any revisions to the manuscript were marked up using the «Track Changes» function. Please find our detailed response below:

Reviewer #2

In the manuscript entitled "Functionalization of Carbon Nanotubes Surface by Aryl Groups: A Review" Alferov et al. review the methods of introducing aryl functional groups onto CNTs surface mainly by means of Gomberg-Bachmann and Billups reactions, and the decomposition of peroxides. I believe that this review article is suitable for publication in Nanomaterials Journal after some revisions as indicated in the comments below.

  1. line 60

replace "comonly" with commonly

RESPONSE 1:

Corrected

  1. line 63

Please, define "SAS"

RESPONSE 2:

Replaced with SDS

  1. line 65-66

"Thus, surfactants interfere during the enzyme immobilization on the nanotube surface."

Why do the authors talk about an enzyme? please clarify

RESPONSE 3:

Enzyme immobilization is a particular case, so a correction was made to the field of electrochemistry and bioelectrocatalysis.

  1. line 77

repleace "substitute" with substituted

RESPONSE 4:

Corrected

  1. Figure 3

Check the accuracy of the reaction mechanism depicted in Figure 3.

RESPONSE 5:

The reaction mechanism has been revised.

  1. Line 128

"Homberg arylation" should be Gomberg arylation. This typos is repeated throughout the manuscript, check carefully.

RESPONSE 6:

Corrected

  1. line 159

repleace "DMFA" with DMF.

RESPONSE 7:

Corrected

  1. line 209

Replace "paradiodbenzene and paradiodbiphenyl" with paradiodobenzene and paradiodobiphenyl

RESPONSE 8:

Corrected

  1. Line 235

"Bilapse" should be Billups

RESPONSE 9:

Corrected

  1. Table 1

Please, check the correspondence of the references cited in the table 1. Maybe there was an error in the numbering.

RESPONSE 10:

The numbering has been corrected

  1. Figure 7

Check the accuracy of the reaction mechanism depicted in Figure 7.

RESPONSE 11:

The reaction mechanism has been revised.

  1. Line 305

Replace "DDS" with SDS

RESPONSE 12:

Corrected

  1. Table 2

Please, check the correspondence of the references cited in the table 2. Maybe there was an error in the numbering.

RESPONSE 13:

The numbering has been corrected

Reviewer 3 Report

This manuscript by Sergey Alferov et al. reviews the aryl functionalization of carbon nanotubes. They introduced the methods of aryl functionalization of carbon nanotubes, their reaction mechanisms, the advantages, and disadvantages of some of the mentioned methods, and the properties of the aryl functionalized products. Finally, they summarized the applications. I think the manuscript is interesting in the field of carbon nanotube modification and application. Overall, the manuscript is well organized, I support the publication on Nanomaterials after considering the following revision suggestion:

I think the authors may compile one table, in which the reaction types are listed with their possible reaction mechanisms (including the problems in mechanism interpretation), advantages, disadvantages, properties of the aryl functionalized carbon nanotubes and preferred application potentials.

non

Author Response

Manuscript ID: nanomaterials-2390043

Type of manuscript: Review

Title: Functionalization of Carbon Nanotubes Surface by Aryl Groups: A Review.

Authors: Pavel Oskin, Iraida Demkina, Elena Dmitrieva, Sergey Alferov *

Received: 24 April 2023

Dear Reviewer!

We would like to thank you for your constructive evaluation of our manuscript. We have completed a revision of the manuscript and have addressed all of the comments/issues raised by each Reviewer.

Any revisions to the manuscript were marked up using the «Track Changes» function. Please find our detailed response below:

Reviewer #3

This manuscript by Sergey Alferov et al. reviews the aryl functionalization of carbon nanotubes. They introduced the methods of aryl functionalization of carbon nanotubes, their reaction mechanisms, the advantages, and disadvantages of some of the mentioned methods, and the properties of the aryl functionalized products. Finally, they summarized the applications. I think the manuscript is interesting in the field of carbon nanotube modification and application. Overall, the manuscript is well organized, I support the publication on Nanomaterials after considering the following revision suggestion:

  1. I think the authors may compile one table, in which the reaction types are listed with their possible reaction mechanisms (including the problems in mechanism interpretation), advantages, disadvantages, properties of the aryl functionalized carbon nanotubes and preferred application potentials

RESPONSE 1:

Table 3 has been added to the section 3 summarizing the results and application areas of various methods of CNT surface functionalization.
